# Protein Adsorption Performance of a Novel Functionalized Cellulose-Based Polymer

**DOI:** 10.3390/polym14235122

**Published:** 2022-11-24

**Authors:** Diana Gago, Marta C. Corvo, Ricardo Chagas, Luísa M. Ferreira, Isabel Coelhoso

**Affiliations:** 1LAQV-REQUIMTE, Chemistry Department, NOVA School of Science and Technology, NOVA University Lisbon, 2829-516 Caparica, Portugal; 2i3N/Cenimat, Materials Science Department, NOVA School of Science and Technology, NOVA University Lisbon, 2829-516 Caparica, Portugal; 3Food4Sustainability—Associação para a Inovação no Alimento Sustentável, Centro Empresarial de Idanha-a-Nova, Zona Industrial, 6060-182 Idanha-a-Nova, Portugal

**Keywords:** adsorption isotherms, adsorption kinetics, dicarboxymethyl cellulose, protein adsorption, cytochrome C, lysozyme

## Abstract

Dicarboxymethyl cellulose (DCMC) was synthesized and tested for protein adsorption. The prepared polymer was characterized by inductively coupled plasma atomic emission spectrometry (ICP-AES), attenuated total reflection Fourier-transform infrared spectroscopy (ATR-FTIR) and solid state nuclear magnetic resonance (ssNMR) to confirm the functionalization of cellulose. This work shows that protein adsorption onto DCMC is charge dependent. The polymer adsorbs positively charged proteins, cytochrome C and lysozyme, with adsorption capacities of 851 and 571 mg g^−1^, respectively. In both experiments, the adsorption process follows the Langmuir adsorption isotherm. The adsorption kinetics by DCMC is well described by the pseudo second-order model, and adsorption equilibrium was reached within 90 min. Moreover, DCMC was successfully reused for five consecutive adsorption–desorption cycles, without compromising the removal efficiency (98–99%).

## 1. Introduction

Proteins are important macromolecules involved in complex physiological processes [1,2]. Protein adsorption, namely adsorptive immobilization of proteins, has been extensively studied and is used in several research fields [3,4,5]. This is a mild immobilization method, providing an easy and fast process for protein capture [4]. Surface properties, protein properties and external conditions are key factors that can affect protein adsorption [2,6,7]. Adsorption on carrier materials focuses on the interactions on both surface and proteins [8]. This process is highly dependent on the isoelectric point of the proteins as well as on the solution pH of the experiments [9]. Understanding the challenges of this bioseparation technique is vital to achieving high-performance separations and favorable adsorption capacities. The design of new efficient biomaterials for protein adsorption should incorporate the influence of physicochemical properties on protein–protein and protein–surface interactions [10,11,12]. Natural materials have particular protein binding positions which can prove useful for this application [11].

Cellulose is a natural polymer abundantly present on the Earth [13]. This linear polysaccharide consists of anhydroglucose units connected by β-1,4-glycosidic bonds [8,14,15,16]. Most cellulosic materials are characterized by biodegradability, biocompatibility and nontoxicity [17]. These materials can be chemically modified, providing promising alternatives for versatile applications [18,19,20,21,22]. The primary functionalization pathway is achieved by the substitution of the hydroxyl groups after cellulose mercerization [23]. Cellulosic materials have been extensively tested as adsorbents, namely for dye adsorption and removal of heavy metals [23,24,25,26,27]. The use of this renewable polymer as a resource for the development of new cellulose derivatives is expected to provide more sustainable alternatives to conventional synthetic polymers [28].

DCMC is a cellulose-based polymer recently developed by this research group [29,30]. This polymer is produced by heterogeneous etherification of cellulose with a β-halomalonate under mild conditions. Its degree of substitution (DS), similarly to other cellulose derivatives such as carboxymethyl cellulose, will influence water solubility and its adsorption performance. An increasing number of carboxylate groups provides additional adsorption binding sites; however, it also increases water solubility [30]. Therefore, a compromise must be achieved to produce a suitable water-insoluble polymer with high adsorption capacity. DCMC is a promising adsorbent that has already been tested for different applications, including dye removal [31] and white wine clarification [30,32].

In this work, we investigate the use of DCMC for protein adsorption, under controlled conditions, for the first time. The adsorption of four model proteins, cytochrome C (Cyt C), lysozyme (Lys), α-lactalbumin (α-LA) and bovine serum albumin (BSA) was studied. Cyt C is a globular heme protein that plays an important role in electron transport chains [9]. Lys is an anti-inflammatory enzyme, used in the food and pharmaceutical industry [33,34]. α-LA is a globular protein that can be used as an additive in food products [35]. BSA is commonly used as a model protein with value for several applications, such as drug carrier systems [36]. Cyt C, Lys and α-LA are small proteins, with comparable size (12–14 kDa), and BSA is a larger protein with 67 kDa. Many experimental studies have been conducted on the adsorption of Cyt C [37,38,39,40,41,42,43], Lys [33,34,44,45,46], α-LA [35,47,48,49,50] and BSA [36,47,51,52] on different materials. The use of different model proteins allows for studying the selectivity of the adsorption behavior of this polymer, specifically the influence of protein net charge and size [53]. The role of electrostatic forces focuses on the interactions between surface charge and protein charge. The model proteins have distinct isoelectric points and, therefore, their overall charge can be tweaked by manipulating the solution pH. It can be then established whether electrostatic attraction or repulsion occurs between the selected proteins and the adsorbent. A better understanding of the protein adsorption process by DCMC will be achieved through modeling analysis of the adsorption isotherms and kinetics. The reusability of the polymer is also an important factor in protein separation since it directly impacts the operation costs [1]. The reusability of DCMC was evaluated through adsorption and desorption cycles using sodium chloride, which is abundant and cheap.

## 2. Materials and Methods

### 2.1. Materials

Malonic acid (99% purity), anhydrous sodium carbonate and potassium iodide at commercial grade were purchased from Panreac (Barcelona, Spain). Trichloroisocyanuric acid (95% purity) was purchased from TCI Chemicals (Zwijndrecht, Belgium). Food-grade cellulose was obtained from ESSECO (San Martino di Trecate, Italy). Sodium hydroxide (97% purity), sodium phosphate monobasic (99% purity), sodium citrate, Cyt C, Lys, α-LA and BSA were purchased from Sigma Aldrich (Darmstadt, Germany). Methanol (ACS reagent), isopropanol (ACS reagent) and acetic acid (glacial) were purchased from Supelco (Bellefonte, PA, USA). Citric acid was obtained from BDH Chemicals (Poole, UK). Sodium phosphate dibasic dodecahydrate was purchased from José M. Vaz Pereira (Lisbon, Portugal).

### 2.2. Preparation of Dicarboxymethyl Cellulose

DCMC was prepared following a procedure previously reported [30]. Briefly, the preparation of DCMC includes the mercerization of cellulose in a sodium hydroxide solution followed by etherification with one molar equivalent of sodium chloromalonate (Figure 1).

### 2.3. Characterization of Dicarboxymethyl Cellulose

FTIR spectra of DCMC were recorded on a Perkin-Elmer FT-IR Spectrometer Spectrum Two (Waltham, MA, USA), equipped with an attenuated total reflection (ATR) cell, in the range of 4000–400 cm^−1^. Before this characterization, DCMC was purified by dialysis against deionized water to remove the remaining salts and small molecules left over from the preparation of the polymer.

Solid state ^13^C MAS NMR spectra of DCMC were acquired in an 11.7 T (500 MHz) AVANCE III Bruker spectrometer operating at 125 MHz (^13^C) and equipped with a BBO probe head. The samples were spun at the magic angle at a frequency of 5 kHz, using 4 mm diameter rotors at room temperature. The ^13^C MAS NMR experiments were acquired with proton cross-polarization and total suppression of sidebands (CP-TOSS) with a contact time of 2.0 ms, a recycle delay of 5.0 s and a sweep width of 37 kHz. Data processing was performed with Topspin 4.1.4 (Bruker, Billerica, MA, USA).

The thermal stability and degradation behavior of DCMC were studied using a Thermogravimetric Analyzer Setaram Labsys EVO (Redon, France). A sample of approximately 15 mg was heated to 500 °C under an argon atmosphere with a heating rate of 10 °C min^−1^.

The pore size distribution was determined through nitrogen adsorption–desorption experiments at 196 °C (77 K) with a Micromeritics ASAP 2010 instrument (Micromeritics, Norcross, GA, USA). The Brunauer–Emmett–Teller (BET) method was used to calculate specific surface area.

The average number of carboxymethyl groups in the cellulose chain is a determining factor in the functionalization of cellulose. The introduction of these new functional groups is determined by the sodium content of its carboxylate groups. DCMC was purified by dialysis against deionized water. Dry dialyzed samples were hydrolyzed by adding 2 mL of nitric acid to a known mass of the polymer (approximately 4 mg). Then, they were incubated at 70 °C for 1 h and analyzed by ICP-AES in a Horiba Jobin-Yvon Ultima model equipped with a 40.68 MHz RF generator, a Czerny–Turner monochromator with 1.00 m (sequential), and an autosampler AS500 (Horiba, Kyoto, Japan). The DS is calculated from the sodium content in samples, based on a procedure previously described in the literature [29].

### 2.4. Protein Adsorption Studies

The adsorption efficiency of DCMC was evaluated with Cyt C, Lys, α-LA and BSA. The proteins were dissolved in phosphate buffer solutions (1 mM, pH 7) with concentrations between 0.04 and 1.5 g L^−1^. Table 1 compares the selected properties of these proteins.

Batch experiments were carried out by adding protein solutions to the polymer. The samples were placed in an orbital shaker (400 rpm) at 25 °C. UV/Vis absorption spectra of all stock solutions were obtained to determine the wavelength of the absorption peaks (Table 1). Initial and equilibrium protein concentrations were determined using a calibration curve. Aliquots of the experiments were collected and assayed spectrophotometrically in a VWR M4 (Ismaning, Germany) to determine variations in protein concentration. The amount of protein adsorbed by DCMC was calculated using the following equation [25]:(1)$$q=\frac{\left(C_{0}-C_{e}\right)}{m}V,$$
where *q* (mg g^−1^) is adsorption capacity; *C*_0_ and *C_e_* (mg L^−1^) are the initial and equilibrium concentrations of protein in the solution, respectively; *V* (L) is the solution volume; and *m* (g) is the adsorbent mass.

The effect of adsorbent dosage was studied to determine the DCMC dosage for the remaining experiments. This experiment was performed by adding polymer doses between 0.25 and 1.5 g L^−1^ to 5 mL of 100 mg L^−1^ of Cyt C solutions. The samples were stirred for 2 h at 25 °C.

Protein adsorption onto DCMC was evaluated through batch adsorption experiments and modelling analysis. A fixed dose of DCMC was placed in contact with protein solutions ranging from 100 to 1500 mg L^−1^. The samples were stirred for 4 h at room temperature. The adsorption data were fitted with adsorption isotherm models, including those of Langmuir and Freundlich [56,57].

The Langmuir isotherm is described by the following equation:(2)$$q=\frac{q_{m}K_{L}C_{e}}{1+K_{L}C_{e}},$$
where *q_m_* (mg g^−1^) is the maximum adsorption capacity, *C_e_* (mg L^−1^) is the equilibrium concentration and *K_L_* (L mg^−1^) is the Langmuir adsorption equilibrium constant, representing the affinity between adsorbate and binding sites.

The Freundlich mathematical model can be written as follows:(3)$$q=K_{F}C_{e}^{1/n}\;$$
where *K_F_* (L mg^−1^) is the Freundlich constant, which relates to adsorption capacity, *C_e_* (mg L^−1^) is the equilibrium concentration and *n* is the heterogeneity factor.

Kinetic adsorption studies were performed following the procedures described previously [31]. In brief, 50 mg of DCMC were placed in contact with 50 mL of protein feed solutions (100 mg L^−1^) at 25 °C. Aliquots of the supernatant were analyzed by spectrophotometry at the appropriate wavelength. These experimental data were fitted to pseudo first-order and pseudo second-order models [58,59].

The pseudo first-order model is expressed by the following equation:(4)$$q=q_{m}\left(1-e^{-K_{1}t}\right),$$
where *q_m_* (mg g^−1^) is the maximum adsorption capacity, *K*_1_ (min^−1^) is the rate constant and *t* (min) is time.

The pseudo second-order model is described as follows:(5)$$q=\frac{q_{m}^{2}K_{2}t}{1+q_{m}K_{2}t},$$
where *q_m_* (mg g^−1^) is maximum adsorption capacity, *K*_2_ (mg g^−1^ min^−1^) is the rate constant and *t* (min) is time.

All the experimental data were analyzed and fitted with the OriginPro 2018 software (OriginLab, Northampton, MA, USA).

To evaluate the fitting of the nonlinear isotherm and kinetic models, error functions were taken into consideration. The correlation coefficient (R^2^) and the sum of the square of the errors were obtained directly from the software. The chi-square (χ^2^) (Equation (6)) test is the sum of squared errors of the differences between the experimental and calculated adsorption capacities and the calculated divided by the corresponding calculated adsorption capacity [60,61,62,63]:(6)$$\chi^{2}=\sum \frac{{\left(q_{exp}-q_{cal}\right)}^{2}}{q_{cal}},$$
where *q_exp_* (mg g^−1^) and *q_cal_* (mg g^−1^) are the experimental and calculated adsorption capacities, respectively.

### 2.5. Reusability Study

To study the reusability potential of DCMC, desorption experiments were performed with Cyt C solutions. The polymer was used for five consecutive adsorption–desorption cycles. In between, DCMC was separated from the solutions by centrifugation. The batch experiments included the adsorption of a 100 mg L^−1^ of Cyt C solution for 2 h, which was followed by desorption with 1 M of NaCl for the same amount of time. Protein removal was calculated by the following equation [64]:(7)$$Protein\;removal\;\left(\%\right)=\frac{\left(C_{0}-C_{e}\right)}{C_{0}}100,$$
where *C_0_* and *C_e_* (mg L^−1^) are the initial and equilibrium protein concentrations, respectively.

## 3. Results and Discussion

### 3.1. Characterization of Dicarboxymethyl Cellulose

#### 3.1.1. FT-IR Analysis

The functionalization of cellulose was characterized by ATR-FTIR spectroscopy. Figure 2 shows the spectra of cellulose (starting material) and DCMC. A characteristic peak at 1634 cm^−1^ confirms the presence of carboxylate groups. Therefore, the cellulose was successfully functionalized. A comprehensive assignment of the absorption peaks to the respective functional groups has been previously described by Gago, Chagas and Ferreira [30].

#### 3.1.2. Solid-State ^13^C NMR Spectroscopy

A chemical shift resonating at 178 ppm was observed in the DCMC ^13^C CP-TOSS spectrum. This shift clearly identifies the presence of the carbonyl moiety from the carboxylate groups (Figure 3). The narrower peaks observed for DCMC in comparison to MCC can possibly be explained by a different relaxation behavior, likely due to a lower degree of polymerization in the functionalized materials.

#### 3.1.3. Thermal Analysis

The thermal degradation behavior of DCMC was examined by thermogravimetric analysis. The thermal degradation pattern of this polymer is reported here for the first time. Figure 4 shows that the thermal degradation of this polymer is a two-step process. The first step (partial weight loss below 3%) is attributed to the loss of moisture content as weakly adsorbed water molecules bound to the carboxylate groups through polar interactions are evaporated, contributing to the dehydration of the polysaccharide structure [65,66]. The weight loss (approx. 61%) observed in the second degradation stage (between 176.81 and 489.78 °C) is attributed to the degradation of the cellulose backbone and of the carboxyl groups (COO-) [67,68]. DCMC has a comparable degradation profile to carboxymethyl cellulose (CMC). Both polymers have two degradation stages: a first stage associated with dehydration and a second stage associated with decarboxylation and CO_2_ release [69]. However, CMC seems to have a higher thermal resistance (higher weight loss step) starting at 250–300 °C instead of 177 °C in the case of DCMC [70]. This is a result of the thermic instability of malonic acid groups that can decarboxylate easily [71].

#### 3.1.4. BET Surface Analysis

The nitrogen adsorption–desorption study was utilized to evaluate the porosimetry of DCMC. The calculated BET surface area was 3.61 m^2^ g^−1^ which is at the limit of detection for this technique. According to the IUPAC classification of physisorption isotherms [72], the isotherm can be defined as Type II, which is associated with macroporous and nonporous adsorbents. These results are comparable to previously published results performed on a similar DCMC polymer [31].

#### 3.1.5. Degree of Substitution

DCMC was synthesized with 1 molar equivalent of sodium chloromalonate. The sodium content in DCMC was determined by ICP-AES. The polymer had 0.80% sodium and a degree of substitution of 0.03. These results are consistent with those obtained previously [30].

### 3.2. Protein Adsorption Experiments Using Dicarboxymethyl Cellulose

#### Effect of Adsorbent Dosage

The amount of adsorbent dosage plays a key role in the adsorption process. A batch experiment with Cyt C solutions and varying dosages of DCMC was conducted to determine an ideal adsorbent dosage. As can be seen in Figure 5, it was found that adsorption efficiency slightly varied between 88 and 93%. The data obtained show that the amount of protein adsorbed did not vary substantially overall. Ultimately, a dose of 1 g of DCMC per liter of protein solution was selected for the remaining adsorption experiments.

### 3.3. Adsorption Isotherms

Figure 6a shows the results of the adsorption of Cyt C. Adsorption capacity increases with increasing initial protein concentration, and the maximum adsorption capacity is 851 mg g^−1^. The saturation capacity of DCMC for the adsorption of Cyt C is close to 200 mg L^−1^ of protein solution. Figure 6b shows the results for the adsorption of Lys onto DCMC. In this case, the adsorption capacity reaches its maximum value at 571 mg g^−1^, and the saturation capacity is close to 400 mg L^−1^ of protein solution.

With the assumption that adsorption by DCMC depends mainly on electrostatic interactions between the polymer and the proteins, it is expected that adsorption increases with the increasing number of positively charged groups [8]. Moreover, pH plays a key role in adsorption since it directly affects the charge of the molecules [31]. For the purpose of this argument, only positively charged amino acids at pH 7 were considered. Lysine and arginine have isoelectric points above 9 whereas histidine’s is closer to 7 [8,32]. Therefore, the number and distribution of histidine was not taken into consideration. Cyt C has a total of 22 positively charged groups (19 lysines, 2 arginines and 1N-terminal), and Lys has 18 groups (6 lysines, 11 arginines and 1N-terminal) [8]. This information agrees with Figure 6 and is a possible explanation for the difference in the adsorption capacity of the proteins. The higher content of positively charged amino acid groups in Cyt C resulting in a higher charge density may be directly related to the enhanced protein removal when compared to the adsorption of Lys.

Experimental data were fitted to two nonlinear isotherm models, Langmuir and Freundlich. The results from this modelling analysis are presented in Table 2. The results show that the adjusted R^2^ values for the Langmuir modeling of the adsorption of Cyt C and Lys are higher than those obtained with the Freundlich model: 0.845 and 0.800 versus 0.728 and 0.680, respectively. Meanwhile, the χ^2^ values are lower when the Langmuir model is applied when compared to those obtained with the Freundlich isotherm model. This implies that the Langmuir model is more suited to describe the adsorption of these proteins onto DCMC. Therefore, it is assumed that the adsorption mechanism of DCMC consists of a monolayer surface adsorption without interaction between the adsorbed proteins [73,74].

### 3.4. Adsorption Kinetics

Kinetic studies were performed with Cyt C and Lys. Figure 7 shows the experimental data and consequent modeling with pseudo first-order and pseudo second-order kinetic models. Results show that the process was faster for the adsorption of Cyt C than for Lys. The adsorption capacity increased with reaction time, reaching maximum values of 89 and 62 mg g^−1^ for the adsorption of Cyt C and Lys, respectively. The adsorption equilibrium of Cyt C was reached after 40 min, whereas the adsorption of Lys took close to 90 min. The faster adsorption of Cyt C can be attributed to the higher number of available positively charged amino acid groups facilitating binding with the polymer.

The adsorption mechanisms of both proteins were evaluated by applying two nonlinear adsorption kinetic models to the experimental data. Table 3 shows the calculated parameters for the adsorption kinetic modeling analysis. The pseudo second-order model adjusted well (R^2^ = 0.937) to the experimental data of the Lys adsorption. The calculated adsorption capacity was similar to the experimental value (62.0 vs. 64.5 mg g^−1^). In fitting the data for the adsorption of Cyt C, the pseudo first-order and pseudo second-order models had the same R^2^ value. To choose the most adequate model, the adsorption capacity and χ^2^ values were considered. The adsorption capacity calculated by the pseudo second-order model was closer to the experimental value while the χ^2^ was lower than those obtained with the pseudo first-order model. The pseudo second-order rate constant for the adsorption of Cyt was higher than that obtained for Lys, which is consistent with the experimental results. Therefore, the pseudo second order was the model chosen for the data fitting of both experiments. For this reason, it is assumed that chemisorption is the rate-controlling step in protein adsorption by DCMC [75,76].

### 3.5. Reusability Study

Regeneration and reusability are important characteristics of an adsorbent. The ability to reuse DCMC in protein adsorption was studied with a 100 mg L^−1^ of Cyt C protein solution and a fixed dose of 1 g L^−1^ of polymer for 2 h. The solution and adsorbent were placed in contact over five consecutive cycles of adsorption and desorption. A solution of 1 M NaCl was used as the desorbing agent for these experiments. During this experiment, the protein removal remained unchanged, with slight variations (Figure 8). After five cycles, the removal ability of DCMC was stagnant at 98–99%. The results showed support for the reuse of this polymer for at least five consecutive cycles without the adsorption performance being hindered. The successful reusability of this polymer will contribute to a more sustainable and economical process.

### 3.6. Proposed Adsorption Mechanism of DCMC

The adsorption of Cyt C onto DCMC is visualized in Figure 9. Cyt C has a characteristic reddish color. After the doped solution is placed in contact with DCMC, the solution loses its color while DCMC turns from white to red. Adsorption of Lys cannot be observed by the naked eye since it has no color. However, through analytical techniques (UV/Vis spectrophotometry) the adsorption process can be further evaluated.

The FTIR spectra of DCMC were obtained before and after being placed in contact with solutions of Cyt C, Lys, α-LA and BSA. Then, the supernatant was removed, and the samples were dried under vacuum overnight. The resulting spectra are presented in Figure 10. Since there were no apparent changes regarding the spectra of DCMC after contact with the buffer solution, α-LA and BSA, the premise that adsorption did not occur in these conditions is supported. However, after adsorption of the positively charged proteins (Cyt C and Lys), there was a change in the absorption band of the functional group. The characteristic peak at 1615 cm^−1^ broadened (with a new maximum absorption at 1630 cm^−1^), which can be attributed to an overlap of the carbonyl functional groups of the polymer and the proteins. Also, a new absorption band at 1540 cm^−1^ characteristic of protein spectra was present. According to the literature, absorption bands at regions 1600–1700 cm^−1^ and 1500–1600 cm^−1^ can be attributed to the amide I and amide II groups of the proteins’ backbone [77,78,79,80]. Amide I absorbance is associated with C=O and C–N stretching, whereas the amide II results from a combination of the N–H in-plane bending and C–N and C–C stretching [77,79]. The difference in the spectra of the polymer after Cyt C and Lys adsorption confirms the presence of these proteins in the DCMC sample. Overall, the alterations in the FTIR spectra suggest that the adsorption process occurred only between the polymer and the positively charged proteins, Cyt C and Lys.

By choosing proteins with different isoelectric points and sizes, we aimed to investigate if protein adsorption onto DCMC relies solely on surface charge. Figure 11 shows the surface representation of the proteins and their electrostatic potential, obtained at a physiological pH (pH 7.4). As can be seen in the figure, α-LA and BSA display high electron density and, therefore, no positive charge at the protein surface. This resulted in electrostatic repulsion between the proteins and the functional groups of DCMC. Contrarily, depletion of electron density on the representation of both Cyt C and Lys promoted electrostatic interactions with the polymer due to excess positive charge on the protein surface. As previously reported, DCMC is mainly deprotonated at a pH greater than 3 [31]. Based on the isoelectric point of the proteins, at the selected pH for this work (pH 7), Cyt C and Lys were positively charged, whereas α-LA and BSA were negatively charged. Both α-LA (14 kDa) and BSA (67 kDa) were used to assess if size might have an influence on the adsorption process. Batch adsorption experiments were performed where DCMC was added to various protein solutions (Cyt C, Lys, α-LA and BSA). DCMC was tested for the adsorption of α-LA and BSA (up to 800 mg L^−1^), and the results were negligible. This is likely due to the electrostatic repulsion between the negatively charged proteins and the polymer, which indicates that the adsorption process is charge dependent.

### 3.7. Comparison with Other Adsorbents

The adsorption capacity of DCMC for the removal of Cyt C and Lys was compared to other adsorbents reported in the literature (Table 4). For this comparison, only studies with adsorbents that followed the Langmuir isotherm model were selected. The Langmuir equilibrium constant describes the bonding interaction between the proteins, and in this case, and the adsorbent. Typically, a higher constant represents a faster adsorption process [87]. However, even though the DCMC equilibrium constant was lower than for other adsorbents, this did not seem to have negatively impacted the adsorption capacity of the polymer. Results demonstrate that DCMC has high adsorption capacity for both proteins when compared to other adsorbents. This supports the assumption that DCMC may be used in protein adsorption with superior results and in a sustainable way.

## 4. Conclusions

DCMC successfully adsorbed positively charged proteins and did not promote the unspecific adsorption of uncharged proteins under the conditions in this study. The adsorption process relied solely on surface charge and electrostatic interactions between DCMC and the proteins. The adsorption of positively charged proteins, Cyt C and Lys, provided information on the adsorption mechanism and kinetics. Equilibrium isotherm data were best fitted to the Langmuir model, suggesting homogeneous monolayer adsorption. A kinetics study showed that the adsorption process followed the pseudo second-order model. After five consecutive cycles of adsorption–desorption, the polymer continued to achieve close to full protein removal (99%), supporting its reuse as a high-performance adsorbent for sustainable processes.

## Figures and Tables

**Figure 1 polymers-14-05122-f001:**
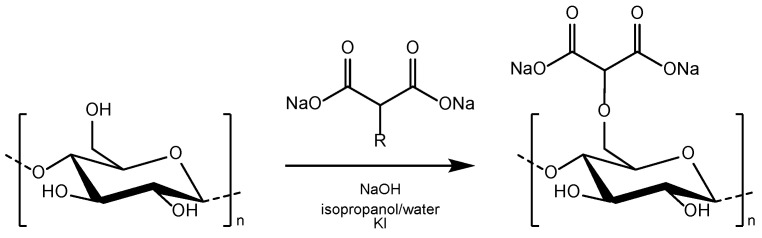
Schematic representation of the synthesis of DCMC. Reproduced from Gago, Chagas and Ferreira [30] (CC BY 4.0).

**Figure 2 polymers-14-05122-f002:**
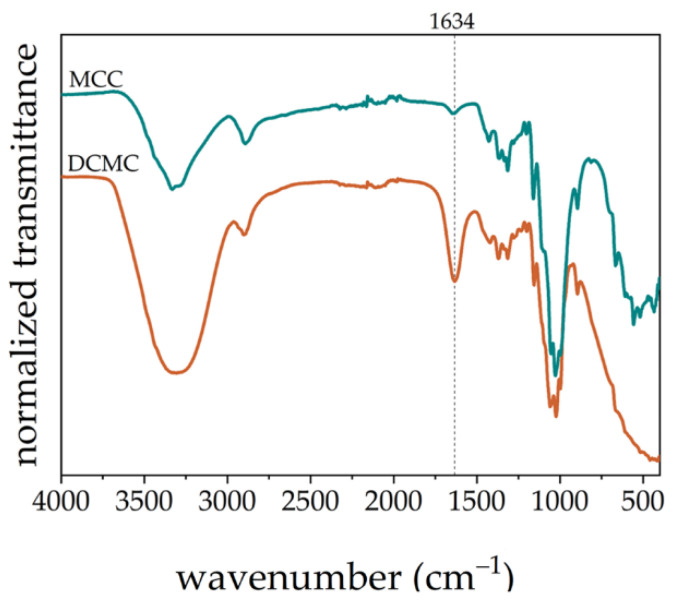
FTIR-ATR spectra of microcrystalline cellulose (MCC) and DCMC.

**Figure 3 polymers-14-05122-f003:**
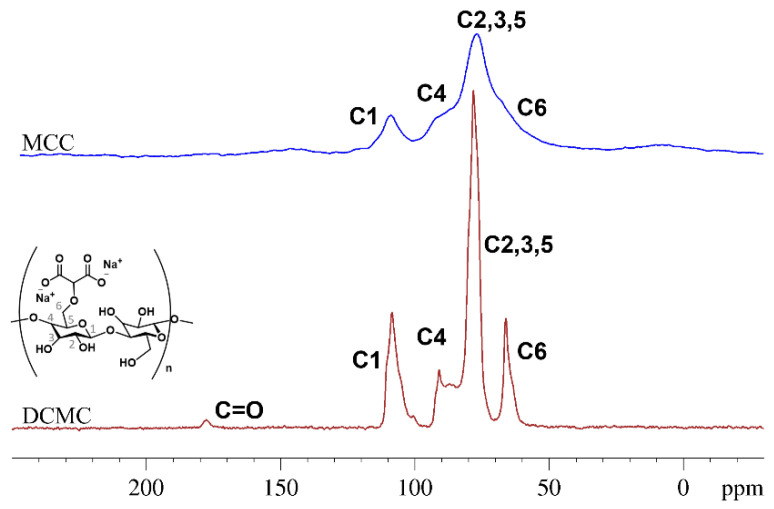
^13^C CP-TOSS NMR spectra of MCC and DCMC.

**Figure 4 polymers-14-05122-f004:**
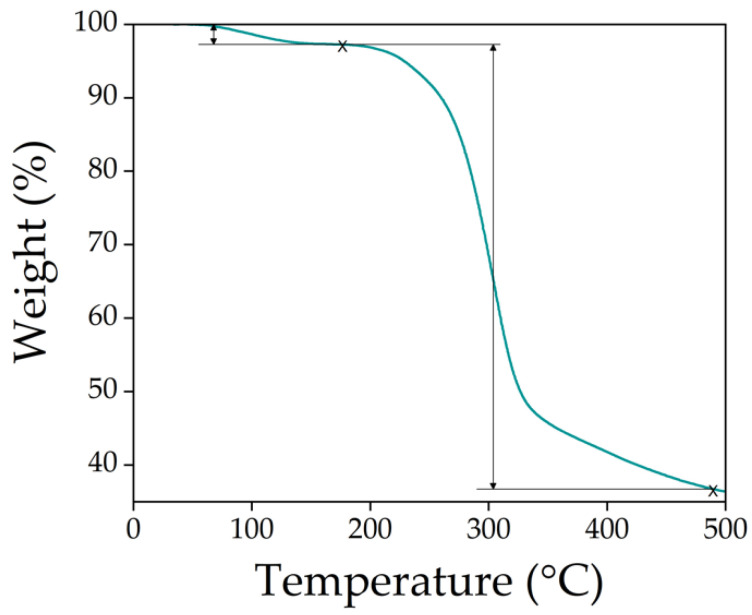
TGA curve of DCMC under argon atmosphere at 10 °C min^−1^ heating rate.

**Figure 5 polymers-14-05122-f005:**
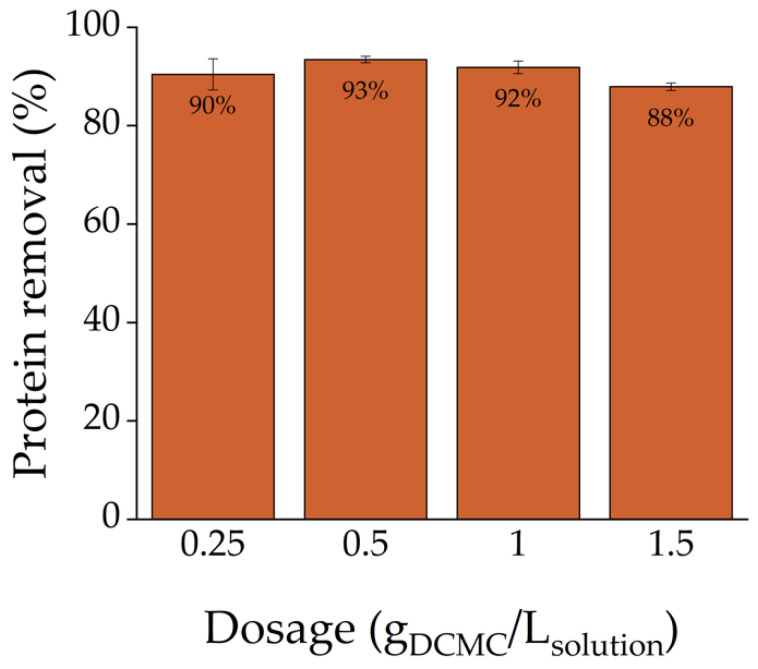
Effect of adsorbent dosage on the adsorption of Cyt C onto DCMC. Each value is reported as mean ± SD of triplicate independent experiments.

**Figure 6 polymers-14-05122-f006:**
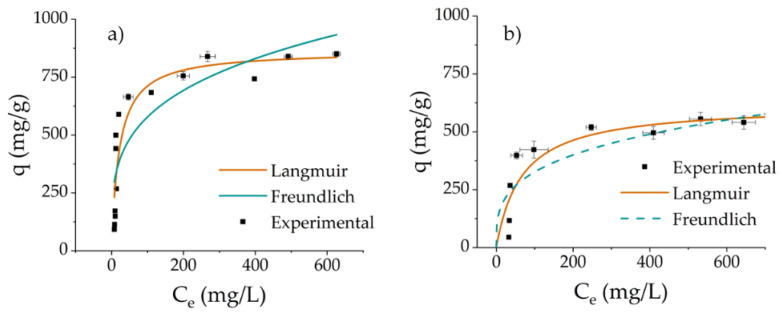
Adsorption isotherm modeling of the adsorption of (**a**) Cyt C and (**b**) Lys onto DCMC. Each value is reported as mean ± SD of triplicate independent experiments.

**Figure 7 polymers-14-05122-f007:**
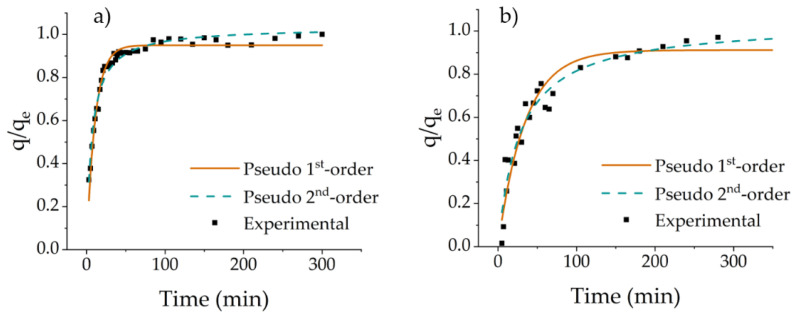
Adsorption kinetics modeling of the adsorption of (**a**) Cyt C and (**b**) Lys onto DCMC.

**Figure 8 polymers-14-05122-f008:**
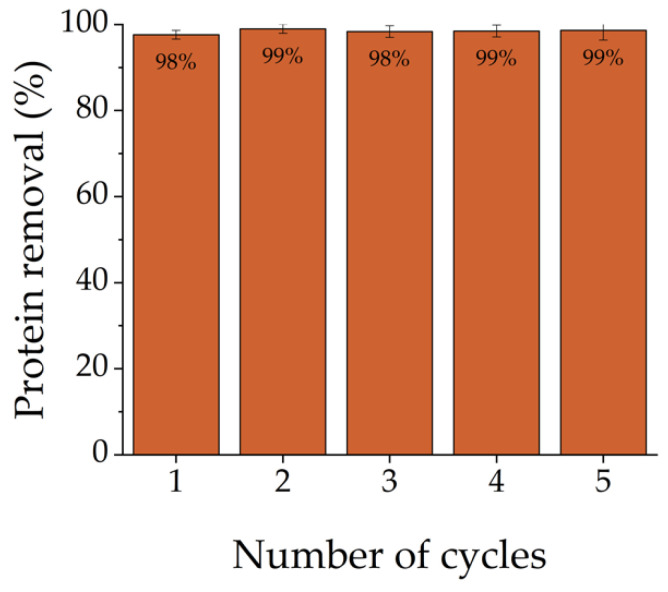
Reusability of DCMC for Cyt C adsorption. Each value is reported as mean ± SD of triplicate independent experiments.

**Figure 9 polymers-14-05122-f009:**
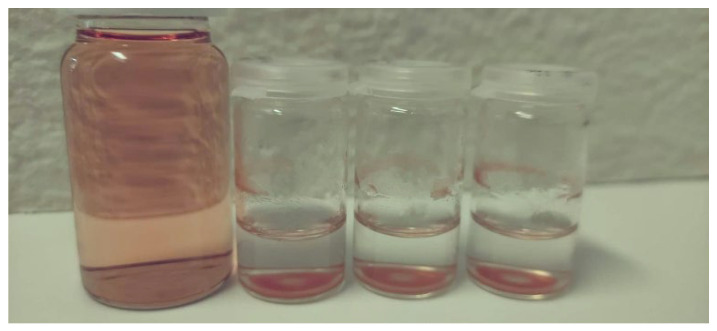
Digital photo of the adsorption of a Cyt C solution onto DCMC.

**Figure 10 polymers-14-05122-f010:**
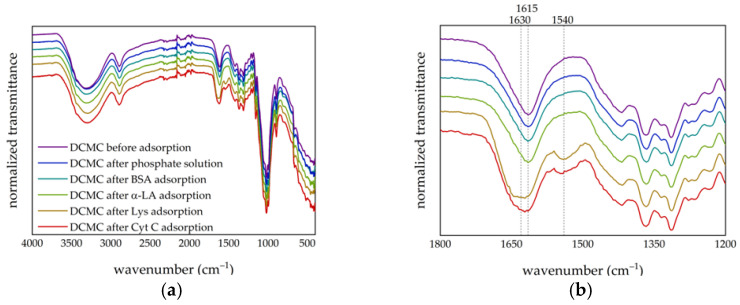
FTIR spectra of (**a**) DCMC before and after adsorption of the solutions (buffer and proteins) and (**b**) zoom of the region between 1200 and 1800 cm^−1^.

**Figure 11 polymers-14-05122-f011:**
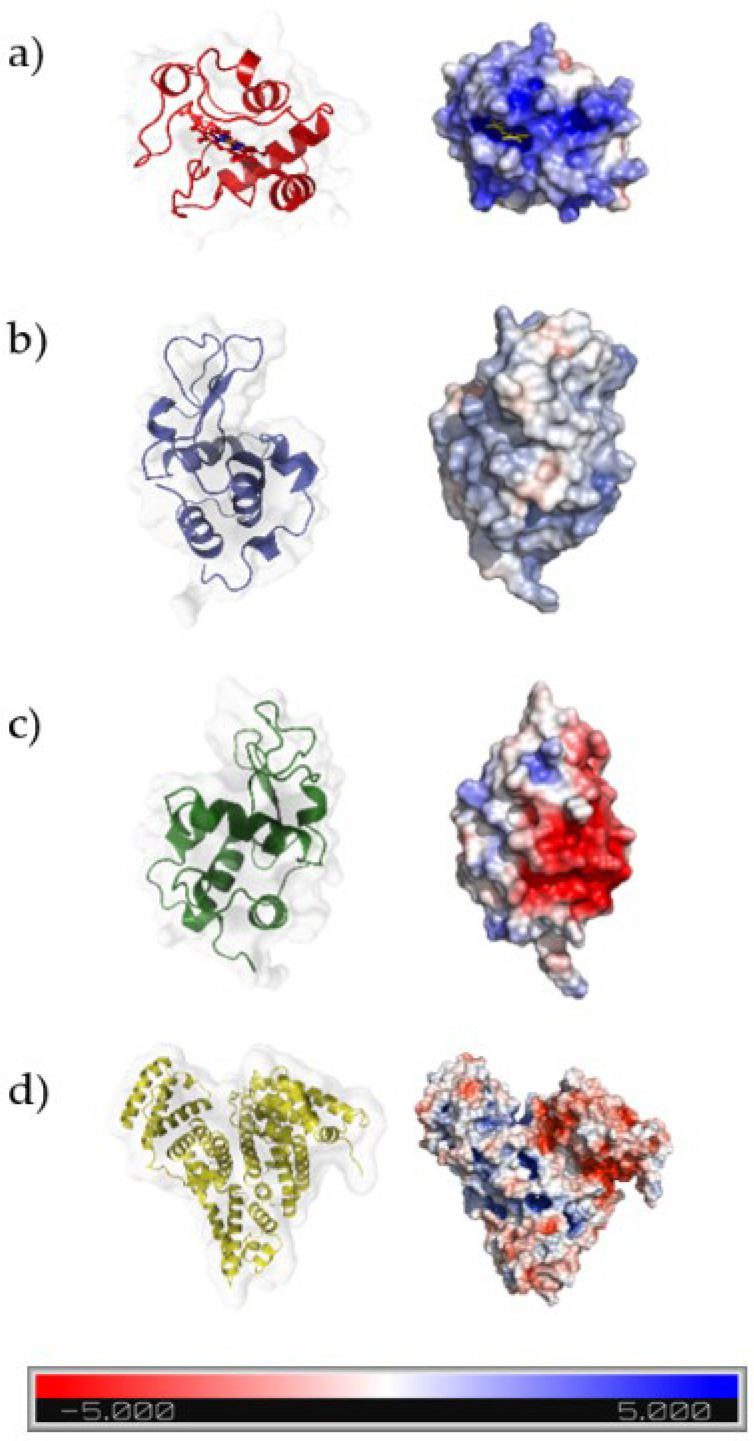
Surface representation, with coloring based on the electrostatic potential of (**a**) Cyt C from horse heart (PDB ID 1HRC; [81]), (**b**) Lys from chicken egg white (PDB ID 5LYZ; [82]), (**c**) α-LA from bovine milk (PDB ID 1F6S; [83]) and (**d**) BSA (PDB ID 4F5S; [84]). Electrostatic potentials were calculated using APBS [85], and images were prepared using Pymol [86].

**Table 1 polymers-14-05122-t001:** Characteristics and properties of tested proteins.

Protein	Isoelectric Point	Molecular Weight (kDa)	λ_max_ (nm)
Cyt C	10–10.5 [54]	12 [54]	410
Lys	10.7 [55]	14 [55]	280
α-LA	4.5 [54]	14 [54]	280
BSA	4.9 [40]	67 [40]	280

**Table 2 polymers-14-05122-t002:** Adsorption isotherms parameters.

Protein	Langmuir				Freundlich			
	*q_m_* (mg g^−1^)	*K_L_* (L mg^−1^)	R^2^	χ^2^	*n*	*K_F_* (L mg^−1^)	R^2^	χ^2^
Cyt C	863.8 ± 57.3	0.047 ± 0.011	0.845	523	3.8 ± 0.7	173.3 ± 44.7	0.728	792
Lys	617.9 ± 58.7	0.015 ± 0.005	0.800	248	3.4 ± 0.9	85.2 ± 38.9	0.680	337

**Table 3 polymers-14-05122-t003:** Adsorption kinetics parameters.

Protein	Pseudo First-Order				Pseudo Second-Order			
	*K*_1_ (min^−1^)	*q_m_* (mg g^−1^)	R^2^	χ^2^	*K*_2_ (mg g^−1^ min^−1^), 10^−3^	*q_m_* (mg g^−1^)	R^2^	χ^2^
Cyt C	0.092 ± 0.003	84.1 ± 0.6	0.969	6	1.600 ± 0.093	91.5 ± 0.8	0.969	4
Lys	0.029 ± 0.003	56.5 ± 1.8	0.914	29	0.563 ± 0.082	64.5 ± 2.2	0.937	23

**Table 4 polymers-14-05122-t004:** Comparison of the maximum adsorption capacities and Langmuir equilibrium constants obtained for the adsorption of Cyt C and Lys onto various adsorbents.

Protein	Adsorbent	*q_m_* (mg g^−1^)	*K_L_* (L mg^−1^)	pH	Temperature (°C)	Reference
Cyt C	DCMC	850.5	0.047 ± 0.011	7	25	This work
	NIMS ^a^	38.61	0.001	7.4	25	[40]
	MIMs ^b^	156.05	0.001	7.4	25	[40]
	PGMA-g-Cell-SO_3_H ^c^	148.58	2.47	9	20	[43]
	PGMA-g-Cell-SO_3_H ^c^	157.13	2.96	9	30	[43]
Lys	DCMC	571.2	0.015 ± 0.005	7	25	This work
	*Navicula* sp.	175.44	0.567	7	30	[50]
	*T. weissflogii*	185.19	0.900	7	30	[50]
	P-EDA-Dye ^d^	588.24	0.006	7	25	[46]

^a^ Nonimprinted mesoporous materials; ^b^ Molecularly imprinted mesoporous materials; ^c^ Sulfonated poly(glycidylmethacrylate)-grafted cellulose; ^d^ Dye-immobilized polyacrylonitrile nanofibers ethylene diamine grafted membrane.

## Data Availability

Not applicable.

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
