# Peer review of "Protein Adsorption Performance of a Novel Functionalized Cellulose-Based Polymer"

_polymers, 2022, doi:10.3390/polym14235122_

Round 1
Reviewer 1 Report
...
The manuscript "Protein adsorption performance of a novel functionalized cellulose-based polymer" by Diana Gago, Marta C. Corvo, Ricardo Chagas, Luísa M. Ferreira, Isabel Coelhoso is a typical work describing special cases of the adsorption of substance A on sorbent B. The work is devoted to detailed writing experiment on the sorption of two proteins on a sorbent of a polymeric nature. But since the actual properties of this polymeric sorbent have not been described or studied (polymer not reliably characterized), and for the purposes of the work and in the conclusion, the manuscripts are not mentioned or discussed, I believe that the authors chose the wrong journal for publication. In my opinion, this is not a profile of Polymers magazine, and I would recommend contacting another physical chemistry magazine.
However, the manuscript is well written but lacks some details of the experiments so that other researchers can replicate the experiments. In addition, the authors incorrectly operate with numerical results - they need to work intensively with the concepts of experimental error.
Below are some private comments on the test.
2 - Headings should not contain word hyphenation.
149 - Experimental conditions are not fully described. In particular, there is no DCMC mass and solution volume.
175 "initial and equilibrium protein concentrations" - How was the initial protein concentration determined?
182 "cellulose has been successfully functionalized" - How did the authors rule out the possibility that the DCMC spectrum is not the sum of the spectra from the original cellulose and impurities of reagents with carboxylate groups?
187 "Fourier-transform infrared spectroscopy (FTIR-ATR) spectra of microcrystalline cellulose (MCC) and dicarboxymethyl cellulose (DCMC)" - Duplication of abbreviations. Abbreviations are introduced once at the first mention, and then only abbreviated forms are used. Check the entire text of the manuscript.
191 - "The polymer had 0.80% sodium and a degree of substitution of 0.03." Again, since the authors did not describe a method for purifying the synthesis product, DCMC, they cannot assert that the sodium they found is a product of a bond with DCMC, and not just an impurity of sodium chloride.
196 "from the carboxylate groups" - Same as "182" - why isn't it a mixture spectrum?
246 - The reviewer does not understand what kind of errors are indicated in the figures. Is this an experimental error? Then how many experiments were carried out? Or is it a method error? If the error of the method (which seems to be the correct approach), then it should be calculated according to (1), and depend on the error in determining the volume, mass, concentrations. Therefore, it is surprising that the errors vary so much for different points. By the way, Ce is also determined with an error, which should be shown in the figure.
246 "863.8±57.3" - What does this entry mean? How many significant digits does the calculated value contain? The authors should carefully familiarize themselves with the concepts of experimental error. The remarks made also apply to Table 3.
246 "523.36" - Are the authors sure that the chi-square test should be given with such accuracy? The remarks made also apply to Table 3.
267 - Give bar-errors of measurements and calculations in the figures.
301 - The table lacks the most important characteristic of the adsorption process - Langmuir adsorption equilibrium constant. Without these data, it makes no sense to compare the results.
311 "for protein adsorption" - Too general conclusion as only 4 proteins were considered in the manuscript and only 2 of them were adsorbed.
...
Reviewer 2 Report
Manuscript ID: polymers-2005707
Diana Gago and co-authors reported "Protein adsorption performance of a novel functionalized cellulose-based polymer". Although the topic is interesting, but some important characterizations were not performed. For possible publication in Polymers, and later better viewership recommended modifucations /additions should be carried out. These comments should be addressed before be possible consideration for publication in worthy Journal of Polymers. I believe it will not take a long for the authors to work on this revision. My comments are,
1. In introduction section 2nd last para starts with, “This work covers the ……. And last para with, “In this work, we investigate……….should be revised and authors work should be in last para only.
2. Preparation of dicarboxymethyl cellulose have not discussed properly. Should be more elaborated for more clarification about procedure for other researchers of relevant field to enhance the readership of this article.
3. Many important characterization were not performed, like SEM, TEM, XRD. Should be performed.
4. Adsorption properties of any adsorbent may not be described without its surface area and pore size determination. BET analysis should be performed for that purpose.
5. Provide origiginal references for Eq 2, 3,4 and 5.
6. In introduction section author claimed four types of proteins (cytochrome C (Cyt C), lysozyme (Lys), α-lactalbumin (α-LA) and bovine serum albumin (BSA)) but adsorption models were applied for only two. Adsorption performance of synthesized Dicarboxymethyl cellulose should be performed.
7. As authors claimed pH is important factor for adsorption of protein. Then why not thet performed the experiment for effect of pH on efficiency of Dicarboxymethyl cellulose?
8. Effect of contact time on adsorption should be performed, for more information and reference Journal of Molecular Liquids 356 (2022) 119036.
9. Proposed mechanism of adsorption of protein onto Dicarboxymethyl cellulose have not discussed properly. Should be discussed in separate sub heading.
10. FTIR analysis after adsorption (used Dicarboxymethyl cellulose) should be carried out
11. Adsorption technique is widely used for the removal on contaminants and purification purposes. Authors should briefly describe adsorption technique and its advantages in introduction section and cite these relevant articles, Microchemical Journal 164 (2021) 105973, Inorganic Chemistry Communications 145 (2022) 110008, Surfaces and Interfaces 34 (2022) 102324.
12. There are so many typo grammatical errors in whole manuscript, like mg/g & mg g-1 , should be revised by some native speaker and formatting should be checked.

Reviewer 3 Report
This manuscript reported preparation of dicarboxymethyl cellulose (DCMC) and its application in protein adsorption. The effects of protein charge and adsorbent concentration on the protein adsorption performance were investigated. However, this paper is not suitable for publication in this form, which needs to be accepted after the revision of the recommendations mentioned below.
Some major concerns are shown as following:
1. Introduction: The introduction part should be restructured since it is considered too short and simple. Some related work should be reviewed in this part. Additionally, the authors are advised to add the reasons for selecting the four proteins as model proteins in the manuscript. Additionally, the most recent literature and some newly developed papers concerning the pollution elimination in industry that may help you are recommended to be added, for instance, Polymers, 2019, 11(11), 1786.; Green Chemistry, 2018, 20(19), 4473-4483. Journal of Cleaner Production, 2020, 259, 120812.
2. The basic information and chemical structure of four model proteins and DCMC should be given.
3. An illustration or a flow chart is recommended to further understand the preparation process of DCMC.
4. In this paper, is the adsorption of protein by DCMC physical adsorption or chemical adsorption? A mechanism illustration is recommended to further understand the adsorption mechanism.
5. The digital photos of the prepared DCMC sample and its adsorption process, and the digital photos of the DCMC sample before and after protein adsorption need to be added to the manuscript.
6. The authors should include the novelty of the manuscript.
7. Why not analyze the adsorbed DCMC? This will help determine the adsorption mechanism. If possible, please add this paragraph to the manuscript.
8. The studies of adsorption thermodynamics should be added to the manuscript.
9. How did authors separate the proteins from DCMC in practical applications?
10. I would recommend the authors to add some tests (such as XPS, TGA, SEM, EDS, hydrophilic and hydrophobic of prepared samples) for the purpose of improving this article.
11. I would recommend to include the economic analysis of this protein adsorption process into the manuscript when the environmental cost was taken under consideration.
12. The adsorption mechanism discussion is very poor, with no further instrumental evidence. The authors need to characterize the material before and after the adsorption, and made conclusions from that result all together with modelling.
13. Please check the entire manuscript for typos and unclear phrases by a native English writer.
Round 2
Reviewer 1 Report
The authors significantly revised the manuscript, significantly improving its quality. Almost all comments were taken into account, so I can only wish the authors further success in their scientific work. The only thing the authors haven't corrected is the error in the constants: "863.8±57.3" should look like "860±60". I hope that the authors will correct these misunderstandings during the final layout.
Reviewer 3 Report
The authors have addrssed most of the comments, so it can be accepted for publication in POLYMERS.